# Experimental Assessment of PA6 Bearing Housing Pressed-Fit for Enhanced Reliability and Multiple Maintenance Process

**DOI:** 10.3390/polym17222971

**Published:** 2025-11-07

**Authors:** Marko Tasić, Žarko Mišković, Radivoje Mitrović, Branislav Đorđević, Aleksandar Dimić, Zoran Stamenić, Lazar Jeremić

**Affiliations:** 1Department Zemun, Academy of Applied Studies Politehnika, 11080 Belgrade, Serbia; 2Faculty of Mechanical Engineering, University of Belgrade, 11060 Belgrade, Serbia; zmiskovic@mas.bg.ac.rs (Ž.M.); rmitrovic@mas.bg.ac.rs (R.M.); adimic@mas.bg.ac.rs (A.D.); zstamenic@mas.bg.ac.rs (Z.S.); 3Innovation Center of Faculty of Mechanical Engineering, 11060 Belgrade, Serbia; brdjordjevic@mas.bg.ac.rs (B.Đ.); ljeremic@mas.bg.ac.rs (L.J.)

**Keywords:** pressed fit, rolling bearing, polymer PA6, press-fitting force, disassembly force, surface roughness, interference fit

## Abstract

This paper presents an experimental method for determining the suitable bore diameter of bearing housings made of polymer designated as PA6, which enables multiple bearing replacement processes. Preceded by analytical calculation, four distinct series of housing samples (each with varying production tolerances) were subjected to testing, where each series comprised three housing samples with identical tolerance specifications. The assembly and disassembly processes of press-fit joints were thoroughly monitored using a force sensor, complemented by equipment for measuring the roughness of contact surfaces. Based on the experimental findings, a recommendation is provided for an appropriate interference fit for the tested bearing housing, providing a suitable solution for multiple maintenance processes. As a summary, the idea of this research is to define the prototype solution for the interference fit of a rolling bearing installed in a PA6 housing. Methods used to examine the proposed solution were surface topography and roundness measuring of PA6 housings, while the press-fitting and dismantling tests of rolling bearings in/from PA6 housings were used to verify it.

## 1. Introduction

Belt conveyor systems are extensively used in open-pit coal mines due to their inherent characteristics of high capacity, robust reliability, and notably, their cost-effectiveness across operational and maintenance cycles. Despite these advantages, a significant amount of the energy dissipates during operating time as a consequence of the dynamic interaction of the conveyor belt and carrying idlers, which highlights potential for energy efficiency improvements. Furthermore, although being manufactured in adherence to current technical standards, numerous existing mechanical assemblies experience failures. Analysis of these assembly failure mechanisms (or their specific components) consistently attributes these failures to either suboptimal geometric features or material properties inadequate for the specified application [1,2,3]. Recent advancements in technology and the development of novel materials provide new opportunities for the manufacturing of machine components. However, it is crucial to note that the application of these new materials does not inherently guarantee enhanced durability from a technical perspective, nor financial viability.

Introducing novel materials into engineering applications necessitates experimental testing (therefore, validation), such as investigations of their mechanical properties [4,5,6,7,8]. The obtained results from these tests provide crucial insights into the material properties defining the limits for its application. Consequently, the validation of a novel material through its characterization inherently leads to an indirect qualification of its corresponding additive manufacturing process for an adequate purpose. It should be emphasized that experimental findings from these tests provide information on the material(s) limitations for specific applications. (One of the critical reviews is summarized in [9]).

Polyamide 6 (in further text, PA6), also known as Nylon 6, is a polymer material often used for the manufacturing of demanding mechanical parts, such as sprockets [10] and gears [11,12]. In general, it can have several different applications, but cast polyamide is well known for its good mechanical characteristics. Furthermore, it possesses enhanced mechanical properties compared to other widely used polymer materials [13]. Its notable chemical resistance is particularly critical for applications in open-pit coal mines, where mechanical components are consistently exposed to environmental effects and working conditions, such as coal dust, moisture, rain, snow, etc. (critical review given in [14,15]). Key properties include favorable tensile strength, elasticity, and toughness. Studies on additively manufactured bearings and their parts made of polymeric materials require particular attention to their performance under actual operating conditions, as well as their manufacturing and assembly characteristics. Petrovic et al. [16] had tested the pressed fit of smaller bearings (with outer diameters of Ø13 and Ø26 mm) and 3D printed housing made of PET-G material, where housing samples were produced in 3 tolerance levels, and assembling and dismantling forces were recorded during these processes. Results had indicated that smaller seat bearing diameters demand higher assembly forces. Velimir et al. [17] tested factors that influence the geometric tolerances of 3D printed bearing housing. Palic et al. [18] used three different sizes of PET-G bearing housings printed in two different tolerance levels and tested the maximum pressed fit force and disassembly force in order to calculate the friction coefficient. You et al. [19] used Finite Elements Analysis (FEA) to predict press-fit curves and maximum mounting force of press-fit assemblies. In their study, they provide a reliable FEA method for predicting the pressing quality during precision press-fit assembly. Wang et al. [20] created an analytical model to calculate press fit force with higher precision, taking into account the influence of non-contact regions of material, while Fiorineschi et al. [21], in their preliminary analysis, had proposed a novel press-fit support mechanism utilizing a three–axis parallel manipulator configured with four actuated legs constrained by cardan and prismatic joints.

In summary, the bearing installation within housings must fully comply with the technical requirements demanded of steel housings. These requirements pertain to:Ensuring a sufficiently high disassembly force for the assembly to prevent axial or tangential relative motion between the housing and the bearing’s outer ring.The aforementioned force must maintain an appropriate value within the expected operating temperature and humidity range.The chemical resistance of the polymer housing material to aggressive environmental agents.The feasibility of multiple bearing replacements within the polymer housing.


The initial press-fit force should not be excessively high to facilitate straightforward assembly and prevent plastic deformation within the housing material. Conversely, following several alternating press-fit and dismantling cycles, the residual force must remain sufficiently high to satisfy the first stated requirement.

Conveyor belt damage represents the most costly failure mode in belt conveyor systems, both in terms of component replacement cost and operational downtime for repair. Integration of polymer housing can significantly delay the onset of such catastrophic failures. This is achieved by allowing the idler to continue rotating around its second installed bearing for a certain period, even if partial melting of the polymer housing occurs, thus preventing an instantaneous shutdown of the entire conveyor system. This leads to an increase in friction between the belt and the steel shell of the idler.

Relying on the previous work of its authors [22], the focus of this study is on a novel idler construction with housings made of polymer PA6. The press-fit joint between these polymer housings and their respective roller bearings must satisfy the same technical requirements as a press fit involving steel housings and bearings. Furthermore, this interference fit must generate a sufficiently high dismantling force to prevent axial or tangential relative motion between the radial bearing and its housing during operational conditions. Crucially, it must also facilitate multiple maintenance cycles, i.e., a repairable feature. In essence, this necessitates that the initial assembly force is not excessively high, yet the force maintained after several maintenance procedures remains sufficient to preclude relative motion between the housing and the radial bearing.

## 2. Materials and Methods

In general, polyamides (PA) are widely considered to possess the most favorable mechanical properties among polymeric materials. Along with the use of standard lithium-based grease for bearings (such as SKF LGMT2 grease [23]—SKF, Goteborg, Sweden), PA6 could be used for application at a temperature of 120 °C (for a shorter period, even up to 160 °C), with a long-term operational time at 100 °C [24] without changes in its mechanical properties., This corresponds to the operational temperature range of rolling element ball bearings. This study investigates the possibility of utilizing polymer PA6 for bearing housing in terms of assessing its capability for multiple maintenance, thus the reliability of the assembly as a whole.

In this research, the following machining parameters for the manufacturing of PA6 housing were used:
depth of cut (ap) of 0.3 mmthe feed rate (f) of 0.2 mm/rev, andcutting speed (vc) of 300 m/min, along air cooling.


These machining parameters are empirical, depending on the cutting tool, the machine characteristics, and the workpiece material, etc. As a consequence of the machining process, the expected surface roughness (Ra) was in the range of 2.5–3.5 µm. These parameters are prescribed by selecting an IT8 tolerance class for the housing’s inner diameter [25]. Additionally, a recommended interference fit value of 0.5 mm is specified for the assembly [26].

Considering the fact that PA6 has a lower elasticity module than steel, it is essential to provide a sufficient interference fit than the one recommended for steel housing-steel bearing pressed fit. Twelve samples of PA6 housing were manufactured in four classes of tolerances of internal diameter: (1) 02, (2) 03, (3) 04, and (4) 05 class (representing 200, 300, 400, and 500 μm of interference fit with bearing). Samples were first manufactured by water jet cutting from 30 mm thick polymer sheets, where ring-shaped parts were cut with an outer diameter of Ø155 mm, while the inner diameter was Ø105 mm. Subsequently, samples were machined on a lathe machine to achieve the required dimensions Ø150/Ø110. Required manufacturing tolerances were achieved by subsequent machining of the inner bore on a lathe machine as well. These 12 housing samples were subjected to experimental testing on a press testing machine for the purpose of determining maximal forces—displacement during 4 cycles of:
(1)rolling bearing press-fit mounting into PA6 housing and(2)dismantling process.


In addition, the aforementioned experiments involve (a) roundness measuring, (b) surface topography investigation, on each of the housing samples. Surface roughness measurements were conducted after each radial bearing press-fit into the PA6 housing and dismantling cycle. This was performed to determine the behavior of surface roughness reduction and to establish an approximate final measure, beyond which subsequent cycles (of press-fit and dismantling) no longer have a significant effect on the interference fit feature.

All housings have an internal diameter of Ø110 mm, an outer diameter of Ø150 mm, and a 30 mm width (Figure 1a). The radial bearing chosen for this testing was SKF 6310-2Z (SKF, Goteborg, Sweden) with a C03 internal clearance tolerance (technical specification was given in [27]). Samples of mounted radial bearings in a PA6 housing are presented in Figure 1b.

Before the experiments, an analytical calculation was conducted to examine the existing interference fit between a steel housing and a steel bearing, which will serve as an input for further investigation on the polymer PA6 as bearing housing material. The preliminary results of this study suggest the feasibility (or suitability) of manufacturing PA6 bearing housings for multiple maintenance cycles of belt conveyor systems. This is directly correlated with the tolerance class of the housing’s inner diameter and the recommended interference fit value necessary for the radial bearing/PA6 housing assembly.

## 3. Analytical Calculation

During the press-fit assembly process, the roughness of both the bearing’s outer surface and the housing’s inner surface undergoes elastic and plastic deformations. Given that bearing steel possesses significantly higher tensile strength than polymer housing, plastic deformations will primarily affect the polymer housing’s surface. These deformations will lead to the deformation of the polymer material’s surface and the alteration of its surface roughness asperities. Upon disassembly, these plastic deformations will persist in the polymer material, and a thin layer of roughness asperities will be removed. Consequently, this will cause a change in diameter tolerances, thereby reducing the interference and the maximum assembly force required for subsequent press-fit operations.

An analytical calculation is performed to analyze the existing pressed fit joint of the steel housing and the steel bearing. The force that loads the pressed fit consists of axial *F_a_
* and radial *F_r_
* components according to Equation (1):(1)$$F=\;\sqrt{F_{a}^{2}+F_{r}^{2}}$$
according to the literature [26], the safety factor of the press fit joint should be in a range 1.1 ≤ *S_µ_
* ≤ 1.25 in order to avoid its critical condition, i.e., joint slipping. It is calculated using Equation (2):(2)$$S_{\mu }=\frac{F_{\mu \;min}}{F},\;S_{\mu }=\frac{F_{\mu }}{F}$$
wherein *F_µ_
* is the friction force achieved on contacting surfaces.

In the inverse case, when the value of the safety factor is set to *S_µ_
* = 1, i.e., when slippage of the press fit joint is allowed, the value of the friction force is obtained:(3)$$S_{\mu }=\frac{F_{\mu }}{F}=1\to F_{\mu }=F=p\times A\times \mu_{k}$$
where

*p*—represents the value of contact pressure*A*—represents pressed fit contact area, with *D* (in mm) as a bearing outer diameter, and *B* (in mm) as a width of the bearing, slightly reduced by the dimension of chamfered edges*µ_k_*—represents the sliding coefficient for pressed fit steel-plastic without lubrication.

If the minimum value of the contact pressure *p_min_
* is placed in Equation (4), the minimum force for the joint to slip is obtained:(4)$$F_{\mu \min}=p_{\min}\times A\times \mu_{k}$$
which corresponds to the minimum pressing force.

The pressed fit contact area is calculated according to Equation (5)(5)$$A=D\times \pi \times B=110\times 3.14\times 27=9330.5{\;\mathrm{mm}}^{2}\;$$


The literature suggests values for the plastic-steel static friction coefficient of *µ_k_
* = 0.33 [26]; however, a value of 0.3 has been adopted for the subsequent calculations.

According to manufacturer recommendations, in this particular case, SKF [27] and the standard for idlers operating in belt conveyors for underground coal mining [28], the tolerance for the interference fit assembly of the SKF 6310 bearing (SKF, Goteborg, Sweden) with *D* = 110 mm and housing is M7/h5. Maximum and minimum values of the tolerance field are given in Table 1. For the recommended housing tolerance, the tolerance band is 35 μm, corresponding to a manufacturing precision class of IT7 and a surface roughness class of N7. Based on these data, a roughness height of polymer housing surface (denoted as *R_p_
*) of 1.6 µm is used in the calculation.

Recommended tolerance band for rolling bearing is 15 μm, with a manufacturing precision class of IT5 and a surface roughness class of N5. Based on these data, a specific roughness height of the bearing steel outer ring surface, denoted as *R_s_
*, equals 0.4 µm, and is used in the calculation.

Furthermore, a roughness peak straightening factor for longitudinal press fits is specified as *φ* = 0.6 [26]. It is used to calculate the loss of interference ∆*P* due to the shearing of the surface roughness asperities and local plastic deformations. *P_max_
* represents the maximum value of the interference fit press and equals 35 µm.

The maximum value of the effective interference fit equals Equation (6):(6)$$P_{\mathrm{efg}}=\left|P_{\max}\right|-\Delta P=\left|P_{\max}\right|-2\varphi \left(R_{p}+R_{s}\right)=0.035-2\times 0.6\left(0.0016+0.0004\right)=0.0326\;\mathrm{mm}$$


Additional values for the calculation of the press fit joint elastic module are given in Equations (7)–(9). Value with index *sh* refers to steel housing, while index *sb* designates steel bearing. Diameters with *s*,*i* in their index represent the steel part internal diameter. It refers to housing diameter in the first equation, and to outer ring race average diameter [24] in the second equation. Diameters with *s*,*o* in their index are the steel part outer diameter. It refers to housing diameter in the first equation, and to bearing outer diameter in the second equation.(7)$$k_{sh}=\frac{1+\frac{d_{s,i}}{d_{s,o}}}{1-\frac{d_{s,i}}{d_{s,o}}}+\nu_{č}=\frac{1+\frac{110}{159}}{1-\frac{110}{159}}+0.3=5.79$$
(8)$$k_{sb}=\frac{1+\frac{d_{s,i}}{d_{s,o}}}{1-\frac{d_{s,i}}{d_{s,o}}}+\nu_{č}=\frac{1+\frac{95.905}{110}}{1-\frac{95.905}{110}}+0.3=14.925$$
(9)$$\frac{1}{E}=\frac{k_{sh}}{E_{s}}+\frac{k_{sb}}{E_{s}}=\frac{5.79+14.925}{2.1\times 10^{5}}=\frac{20.715}{2.1\times 10^{5}}\Rightarrow E=\frac{2.1}{20.715}10^{5}=0.1\cdot 10^{5}{\;N/\mathrm{mm}}^{2}$$


Maximum effective interference fit is:(10)$$\varepsilon_{\max}=\frac{P_{efg}}{d}=\frac{0.0326}{110}=0.0002964$$


By Equation (11), the maximum contact pressure equals:(11)$$p_{\max}=\varepsilon_{\max}\cdot E=0.0002964\times 0.1\times 10^{5}=2.964{\;N/\mathrm{mm}}^{2}$$


According to the preceding calculations, where *µ_p_
* = 0.05 and *µ_s_
* = 0.03 represent the pressing coefficient of friction and static coefficient of friction [26], the maximal pressing and dismantling forces are calculated using Equation (12) and Equation (13). As a justification, a lower value of the static friction coefficient *µ_s_
* was adopted for (higher) safety factor considerations.(12)$$F_{p\;max}=p_{g}\times A\times \mu_{p}=2.964\times 9330.5\times 0.05=1382\;N\to 140\;\mathrm{kg}$$
(13)$$F_{d\;max}=p_{g}\times A\times \mu_{s}=2.964\times 9330.5\times 0.08=2212\;N\to 225\;\mathrm{kg}$$


These two calculated forces, a consequence of the interference tolerance specified by the standard in their gist, represent reference values that must be experimentally validated and slightly exceeded through the interference fit assembly of the polymer housing and the steel bearing following multiple assembly and disassembly cycles. It should be noted that polymers exhibit a higher tendency for creep and changes in their elastic modulus over time. Taking these features into account, the experimental testing is repeated until the measured pressing and dismantling forces converge with those obtained in the previous experimental cycle.

## 4. Testing Methods

### 4.1. Surface Topography Investigation

As a recall, all PA6 housings were manufactured in four classes of tolerances of internal diameter (02, 03, 04, and 05 classes), providing interface fit of 200, 300, 400, and 500 μm with bearing. The machining was carried out according to the machining parameters listed in Section 2. Due to insufficient manufacturing precision, deviations from the specified nominal dimensions were observed.

This investigation began with an examination of the surface of each housing sample made of PA6. The surface analysis was performed using a Hirox KH 7700 digital microscope (HYROX, Tokyo, Japan, 2016). It represents an advanced optical microscope suitable for inspecting object surfaces, measuring features in 2D and 3D, generating 3D profiles, etc. Figure 2 shows different lens magnifications.

Surface roughness testing is conducted on each housing sample before 1st cycle and after 2nd, 3rd, and 4th cycles of radial bearing press-fit mounting into the housing and dismantling process on the pressing test machine. Therefore, 1st cycle of measuring took place before the experiment started, and after each cycle of assembling and dismantling. This was performed to determine the number of cycles after which the surface roughness peak height exhibits no significant reduction compared to the previous cycle. The MahrSurf M440 SD26 equipment (Mahr, Esslingen, Germany, 2016) shown in Figure 3 was utilized for this purpose. The following parameters were selected for this test:Traversing length—17.5 mmCutoff length—2.5 mmMeasuring speed—0.5 mm/sMeasuring interval—1.5 μm


### 4.2. Roundness Measuring

All twelve samples of PA6 housing (with internal Ø110 mm, outer diameter of Ø150 mm, and 30 mm width) were conducted as well for roundness measuring before experimental testing on a press testing machine. As another recall, all housings were manufactured in four groups of three internal diameters each of 109.8, 109.7, 109.6, and 109.5 mm each in the tolerance field of H9 and surface roughness of *R_a_
* = 6.3 μm [25].

### 4.3. Press-Fitting and Dismantling Test of Rolling Bearings in/from PA6 Housings

Since the maximum pressing and dismantling force was initially unknown, and the load capacity of the Mechanical Faculty’s laboratory press testing machine was limited to 10,000 N, the 1st cycle of rolling bearings press-fit mounting into PA6 housings and dismantling test was conducted in an external laboratory. This initial testing utilized an Instron 150 model 5584 kN press testing machine (Instron, Norwood, MA, USA, 1995), whose maximum pressing force significantly exceeded the anticipated values. Data acquisition and analysis are performed using EVOTest Sand Plasm Compress software (Evotest, Inc., CA, USA, https://bpsinstrument.com/filesdirectserver/itp1/z_itp_2058rwkn/EVOTestmanualENz-z3975249232.pdf, accessed on 4 December 2024).

The values for maximum forces and dimensions (including interference fit and average interference fit in class) after 1st cycle of press-fitting and dismantling of the roller bearing into each PA6 housing sample are presented in Table 2.

Based on these results, it was estimated that maximal press forces will not exceed 10 kN in each subsequent cycle of testing. Additionally, it was determined that the experiment could be subsequently performed on the electro-mechanical Tinius Olsen H10KS tensile/press testing machine (Tinius Olsen, Horsham, PA, USA, 2016) depicted in Figure 4a. Its maximal force of 10 kN is utilized for the remaining 3 cycles of press-fitting and dismantling, and it should be emphasized that it was used only as a mechanism with manual control of press force. Force sensor YZC-516C 2t—S type (Guangdong South China Sea Electronic Measuring Technology Company Limited, Mayong, China, 2022) and data acquisition system Arduino UNO R3—ATmega328-16MHz (Arduino, Monza Italy, 2021) were used for controlling the force level. An experimental setup depicting bearing press-fit mounting process into PA6 housing is presented in Figure 4b.

It should be emphasized that the sensor and system, as well, were calibrated on the aforementioned Tinius Olsen H10KS tensile testing machine before the remaining three cycles of testing. This machine is suitable for a variety of tensile and compression tests. The verification was conducted through several test series, with a remark that the equipment demonstrated a satisfactory level of precision.

The equipment was then mounted, followed by manually eliminating the initial gap until the force sensor registered a minimal force. From that point, the manual tightening wheel of a press was rotated with a 1.5 mm step, after which a brief pause followed to stabilize the force reading, and the resulting force increase (or decrease) was recorded. It is crucial to understand that the initial force increase, particularly during the dismantling process, occurs within the elastic deformation domain. This precedes any relative motion between the two interference-fit components. Consequently, in this phase of the experiments, the rotation of the manual tightening wheel does not result in a corresponding relative linear motion of both elements.

## 5. Results and Analysis

This Section begins with the results of the examination of the surface of housing samples made of PA6. No micro-cracks or other imperfections that would adversely affect the manufacturing quality were detected on any of the tested samples. Figure 5a,b shows the representative surfaces of samples 102, 204, 303, and 305. Magnification, along with a scale bar, is given on each surface figure.

In Table 3 are presented the surface roughness results (through parameter *R_a_
*) for each PA6 housing sample.

It could be observed that testing samples designated as 102, 202, and 302 exhibited low initial testing forces, leading to their anticipated rejection. Consequently, only one measurement was performed after the third test cycle for these samples. The remaining samples, with the exception of sample 104, showed uniform values across the remaining three cycles of press-fit mounting and dismantling.

The results of the roundness measurement are given in Table 4. The internal diameter was subjected to multiple measurements across all orientations; nevertheless, to ascertain roundness, only the extreme values obtained along two orthogonal axes are tabulated.

Results shown in Table 2 indicate that the achieved dimensions of the housing bore deviate from the targeted ones due to manufacturing inaccuracy, resulting in the difference between the target interference fit and the average interference fit values in class.

The diagrams shown in Figure 6, Figure 7, Figure 8 and Figure 9 depict force-displacement trends for the 2nd, 3rd, and 4th cycles of the press-fitting and dismantling process of the roller bearing and PA6 housing. The subscript “d” denotes the force variations during the disassembly of the pressed-fit assembly of the roller bearing and the PA6 housing. In this particular case, the measured displacement on each diagram represents the relative position of the roller bearing and the housing at the beginning of the pressing-in/dismantling process (when the displacement equals 0) and at the end of the force measurement process.

The results obtained after the 2nd testing cycle (of press-fitting and dismantling) for housing samples 102, 202, and 302 unequivocally indicated that the acquired data did not meet the required measurements and are significantly below the needed ones necessary for satisfactory interference fit. Another subsequent cycle was conducted exclusively to validate this preliminary observation and tendency, after which a decision was made to exclude these samples from further analysis and testing.

During testing, samples 103, 203, and 303 demonstrated pressing and dismantling forces that were slightly below the targeted one, i.e., calculated in Equations (12) and (13). A noteworthy observation was the unexpectedly low force values recorded during the 2nd testing cycle for sample 103, potentially attributed to lubricant from bearing contamination of the interfacial contacting areas.

The testing results for samples 104, 204, and 304 were found to be acceptable. Although sample 104 exhibited slightly lower measured force values, these were still within the range considered satisfactory compared to the calculated ones in Equations (12) and (13). This is the same sample that exhibited a significant reduction in surface roughness parameters during the surface roughness measurement after the 4th cycle.

The measured forces for test samples 105, 205, and 305 were substantially above the prescribed threshold. Furthermore, after four testing cycles, these samples exhibited a stagnation in the reduction in force values, both during the press-fitting and the dismantling process.

As a general observation, sudden transient ‘jumps’ and ‘drops’ (i.e., fluctuations) in the force values (for instance in Figure 6, partly in Figure 7) can be primarily attributed to insufficient interference fit between rolling bearing and PA6 housing, which is clearly evident in case of PA6 test specimens 102, 202, and 302, and partially in 103. In addition, force ‘drops’ could also be attributed to the non-uniformity of the surface texture. For the remaining specimens, the force increments observed during both press-fitting and dismantling processes were in direct correlation with the recoverable elastic strain. As a recall, it is essential to emphasize that the displacement of the pressing machine was conducted in small intermittent steps every few seconds, i.e., after stabilizing the force value on the sensor. In this context, this provides (as well) a justification for the selection of the static friction coefficient in the preliminary analytical calculation.

## 6. Conclusions

Acquired measurement data indicated a significant reduction in surface roughness peaks caused by the initial first two cycles of the press-fitting / dismantling process. This observation directly correlated with a reduction in both maximum (axial) press-fitting force and the maximum dismantling force. The marginal variance observed between the final two measurements suggested a convergence of the measured parameters, implying that subsequent testing cycles would not yield a substantial further decrease in these values.

The dimensional tolerances of samples 104, 204, and 304 were assessed as being at the lower limit of acceptable manufacturing specifications. Conversely, samples 105, 205, and 305 exhibited performance characteristics that deem suitable for further evaluation under varied operational regimes.

As a summary, the general conclusion that can be drawn is:Polymer PA6 found to be suitable for multiple maintenance processes of bearing/PA6 housing assembly.PA6 housing samples designated as 105, 205, and 305 (with 05 class of tolerances of internal diameter ensuring 500 μm of interference fit with bearing) showed the best results in terms of achieving a sufficiently high assembly and disassembly force after multiple test cycles. Before the 1st test cycle, these specimens achieved an average interference fit with the bearing’s outer ring of 447 μm. The surface roughness before the experiment varied within the range of 7.4–10.6 μm, while after the 4th cycle, it varied from 2.5 to 2.8 μm (Table 3). In addition, the roughness did not significantly differ from that of the other specimens and, as such, did not show a crucial influence on the pressing and disassembly force of the rolling bearing/PA6 housing assembly.The research confirmed the stability and convergence of key mechanical parameters after a limited number of maintenance processes that include press-fitting and dismantling of a rolling bearing / PA6 housing assembly, indicating predictable interface behavior.Finally, a recommendation regarding the tolerance class for the housing’s inner diameter would be IT8, in addition to the recommended interference fit value of 0.5 mm for the assembly.


The results obtained in this study support the premise that it is feasible to manufacture a PA6 housing capable of establishing a robust interference fit with a rolling bearing, even after multiple maintenance cycles. However, it remains necessary to verify all these features under the anticipated service loads of an idler assembly. Therefore, further investigation will focus on the PA6 housing’s capability of generating and maintaining a load-bearing interference fit under relevant operational conditions for idler applications.

## Figures and Tables

**Figure 1 polymers-17-02971-f001:**
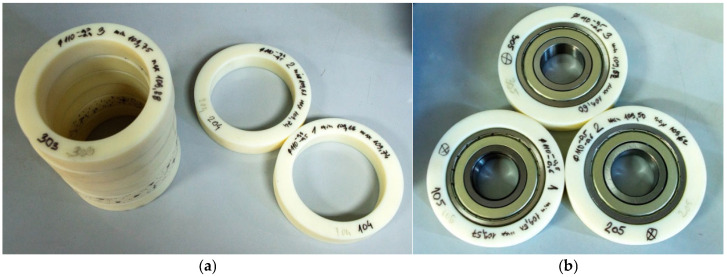
(**a**) 12 samples of 3D printed bearing housing; (**b**) mounted radial bearings SKF 6310-2Z into PA6 housings.

**Figure 2 polymers-17-02971-f002:**
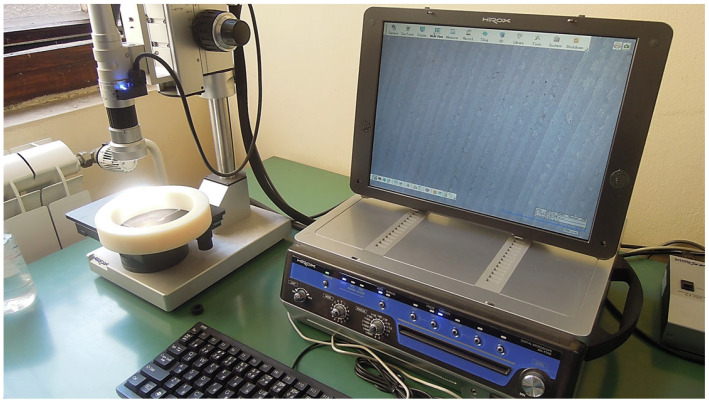
Hirox microscope used for surface analysis of PA6 housing samples.

**Figure 3 polymers-17-02971-f003:**
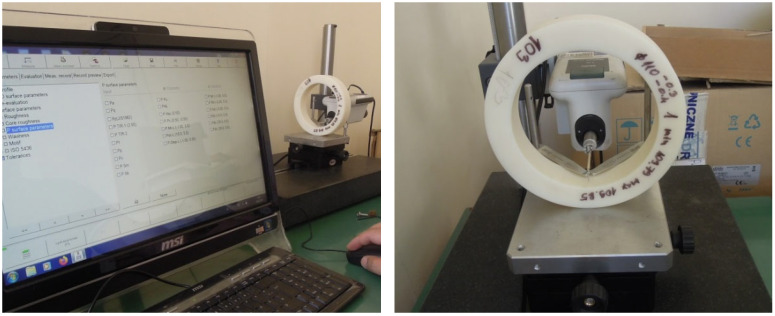
Testing setup for surface roughness testing on MahrSurf M440 SD26 equipment (designation 103 indicates the first sample from class 3 tolerance as an example).

**Figure 4 polymers-17-02971-f004:**
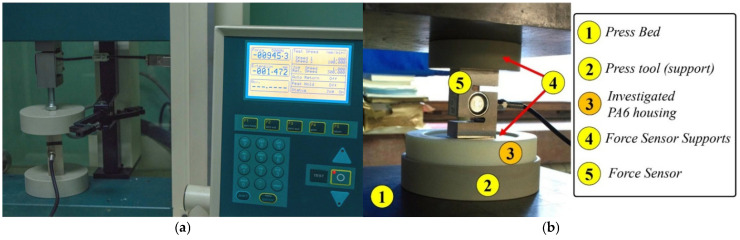
(**a**) Tinius Olsen H10KS tensile testing machine (located at the Faculty of Mechanical Engineering, University of Belgrade) used for remaining three press-fit and dismantling cycles; (**b**) experimental setup.

**Figure 5 polymers-17-02971-f005:**
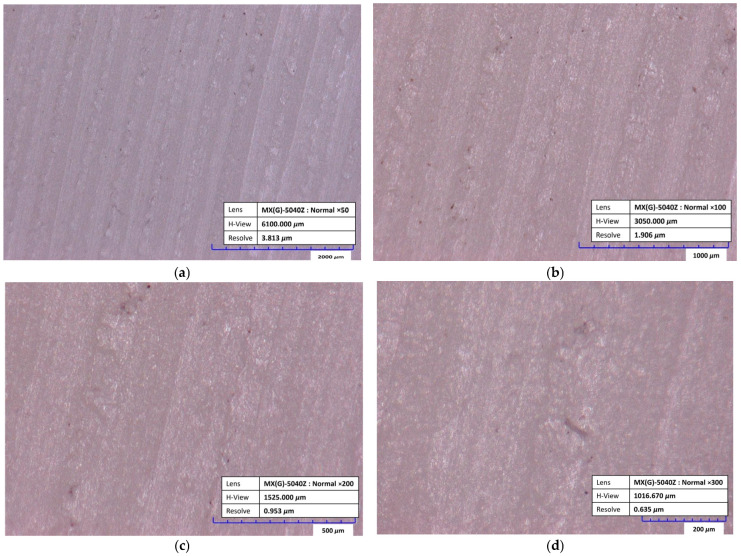
(**a**) surface of 102 sample; (**b**) surface of 204 sample; (**c**) surface of 303 sample; (**d**) surface of 305 sample.

**Figure 6 polymers-17-02971-f006:**
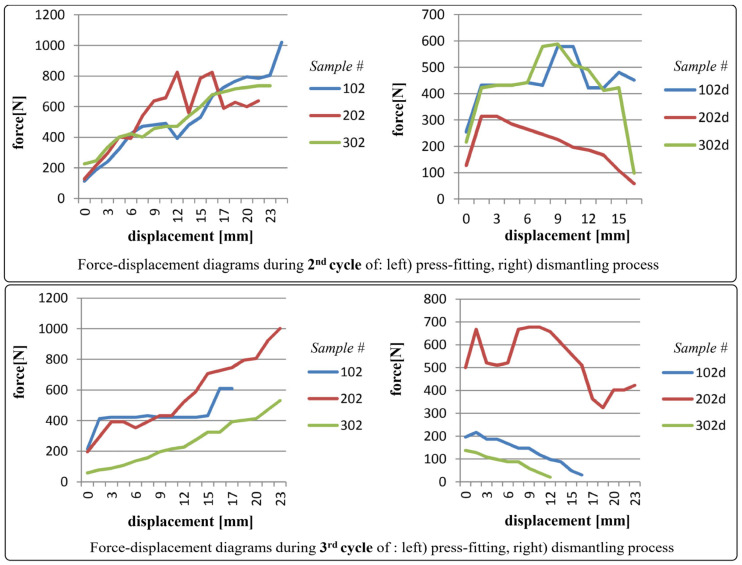
Force-displacement diagrams for samples 102, 202, and 302 during 2nd and 3rd cycles of press-fitting and dismantling process.

**Figure 7 polymers-17-02971-f007:**
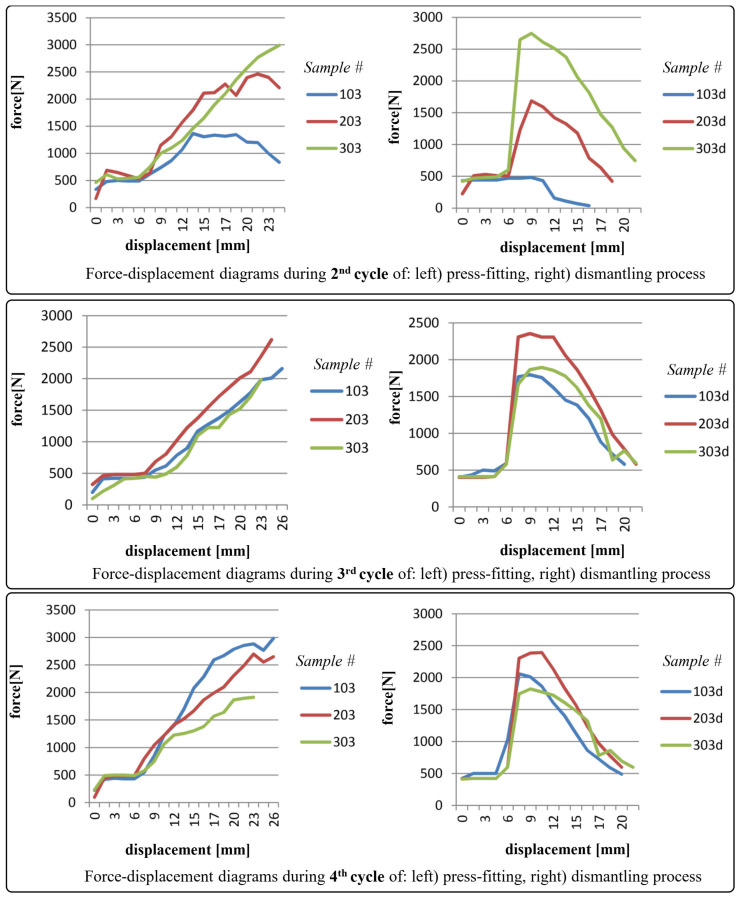
Force-displacement diagrams for samples 103, 203 and 303 during 2nd, 3rd and 4th cycles of press-fitting and dismantling process.

**Figure 8 polymers-17-02971-f008:**
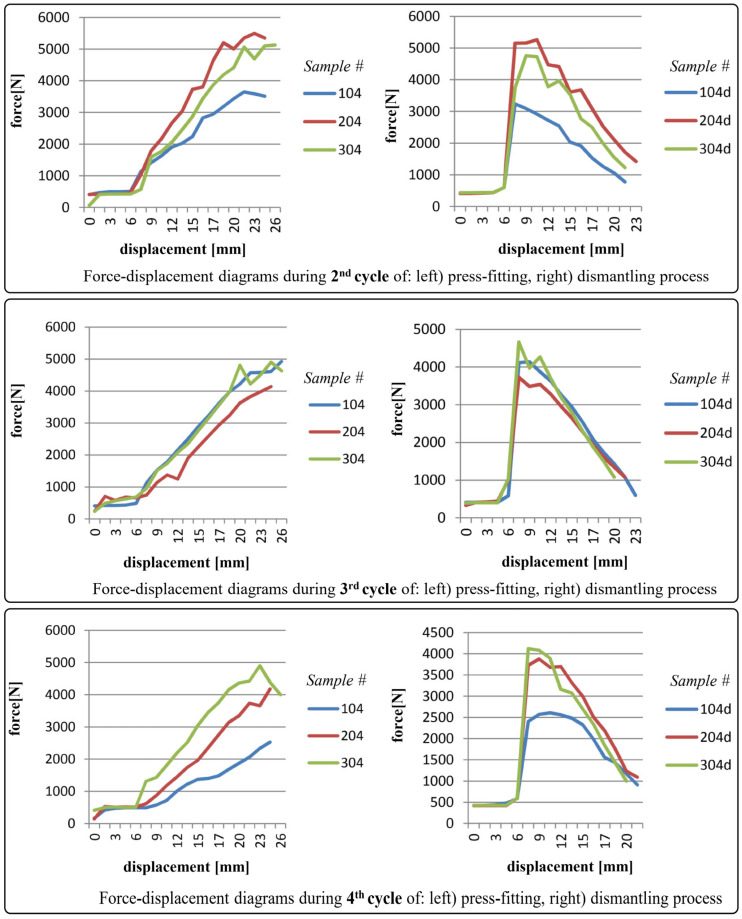
Force-displacement diagrams for samples 104, 204 and 304 during 2nd, 3rd and 4th cycles of press-fitting and dismantling process.

**Figure 9 polymers-17-02971-f009:**
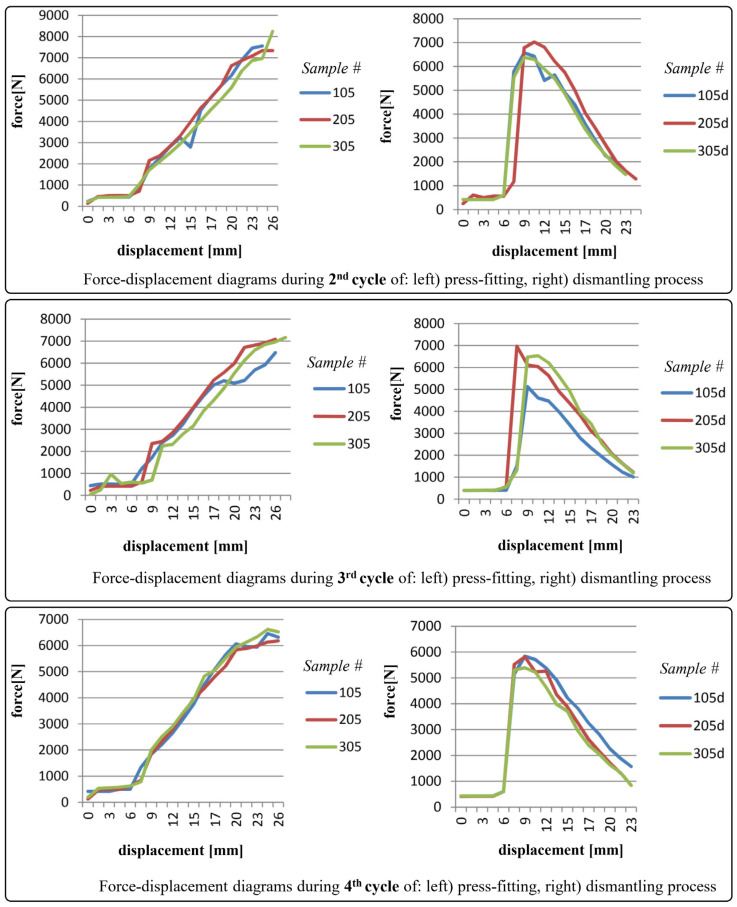
Force-displacement diagrams for samples 105, 205 and 305 during 2nd, 3rd and 4th cycles of press-fitting and dismantling process.

**Table 1 polymers-17-02971-t001:** Maximum and minimum values of tolerance field M7 and h5 (for nominal size Ø110 mm).

Tolerance Field	Max Values [mm]	Min Values [mm]
h5	+0.000	−0.015
M7	+0.000	−0.035

**Table 2 polymers-17-02971-t002:** The maximum forces and dimensions after 1st cycle of press-fitting and dismantling of the roller bearing into each PA6 housing sample (Instron 150 kN press testing machine).

Sample Designation	102	202	302	103	203	303
Maximal press force [N]	8190	9060	8090	10,030	11,189	11,100
Maximal dismantling force [N]	8150	8620	8490	9810	10,400	10,690
Average dimension [mm]	109.98	109.95	109.98	109.91	109.91	109,86
Interference fit [mm]	0.018	0.046	0.014	0.086	0.092	0.136
Average interference fit in class [mm]	0.026			0.105		
Sample designation	104	204	304	105	205	305
Maximal press force [N]	13,020	13,520	11,770	17,810	17,830	17,020
Maximal dismantling force [N]	12,140	12,350	11,350	16,440	16,050	16,680
Average dimension [mm]	109.77	109.68	109.736	109.54	109,56	109.606
Interference fit [mm]	0.234	0.322	0.264	0.460	0.444	0.394
Average interference fit in class [mm]	0.273			0.433		

**Table 3 polymers-17-02971-t003:** Surface roughness *R_a_
* [µm] results for each tested housing sample.

Surface Roughness *R_a_ * [µm]
Sample Designation	Before Testing	After 2nd Cycle	After 3rd Cycle	After 4th Cycle
102	9.723	/	3.201	/
202	10.550	/	2.574	/
302	9.382	/	2.841	/
103	8.238	3.551	3.119	3.313
203	11.158	3.587	3.454	3.388
303	11.114	3.218	3.023	3.168
104	7.921	3.156	2.893	1.814
204	10.720	3.479	3.442	3.338
304	10.386	3.471	3.125	3.128
105	10.602	3.102	2.871	2.741
205	9.662	2.912	2.869	2.687
305	7.428	2.992	2.596	2.576

**Table 4 polymers-17-02971-t004:** Measured dimensions of PA6 housing samples.

Sample Designation	102	202	302	103	203	303
Targeted interference fit [mm] -Dimension [mm]	0.2 109.8–109.852	0.2 109.8–109.852	0.2 109.8–109.852	0.3 109.7–109.752	0.3 109.7–109.752	0.3 109.7–109.752
Maximal dimension [mm]	109.87	109.9	109.92	109.79	109.64	109.75
Minimal dimension [mm]	109.97	110.0	110	109.85	109.78	109.88
Average dimension [mm]	109.92	109.95	109.96	109.82	109.71	109.815
Interference fit [mm]	0.08	0.05	0.04	0.18	0.29	0.185
**Average interference fit in class [mm]**	**0.057**			**0.218**		
**Sample designation**	**104**	**204**	**304**	**105**	**205**	**305**
Targeted interference fit [mm] -Dimension [mm]	0.4 109.6–109.652	0.4 109.6–109.652	0.4 109.6–109.652	0.5 109.5–109.552	0.5 109.5–109.552	0.5 109.5–109.552
Maximal dimension [mm]	109.68	109.61	109.59	109.51	109.50	109.52
Minimal dimension [mm]	109.74	109.74	109.68	109.57	109.62	109.60
Average dimension [mm]	109.71	109.675	109.635	109.54	109.56	109.56
Interference fit [mm]	0.29	0.325	0.365	0.46	0.44	0.44
**Average interference fit in class [mm]**	**0.327**			**0.447**		

## Data Availability

The original contributions presented in this study are included in the article. Further inquiries can be directed to the corresponding author.

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
