# Peer review of "Experimental Assessment of PA6 Bearing Housing Pressed-Fit for Enhanced Reliability and Multiple Maintenance Process"

_polymers, 2025, doi:10.3390/polym17222971_

Round 1

Reviewer 1 Report

Comments and Suggestions for Authors

This article deals with the experimental determination of the bore diameter of metric bearing housing made of PA6 polymer, which allows for multiple bearing replacement processes. In my opinion, the article needs to be methodologically edited and improved, also considering the comments below:

  1. The authors in the Abstract talk about "determining the optimal diameter of the bore of bearing bushings made of a polymer designated as PA6"...
  2. What do the authors consider optimal, what criteria must the optimal diameter meet and what method was used to perform the optimization?
  3. Lines 120 - 125 - On what basis were the specified machining parameters used in the given research? Note: surface roughness is a consequence of the machining parameter settings, it does not belong to the settings.
  4. Lines 115 - 117, similarly 125 - 126, - references to literary sources are missing in the statements
  5. The designations in the text of symbols and variables should be the same as in the equations (slanted, with subscripts, etc ...) e.g. line 187 pmin, line 195 d; but it is necessary to check this in the entire manuscript.
  6. I see a question mark after equation (5), please check.
  7. Justify why static friction coefficient of µk = 0.3 was adopted for the subsequent calculations.
  8. In lines 197 and 198, the authors talk about tolerances of 35 um, but in Table 1 about tolerances with a value of 0.035 um. What technology do the authors want to use to produce a 110 mm diameter component with a tolerance of 0.015 or 0.035 um? Water jet? Below in line 201, the recommended tolerance value is 15 um (this is a big difference compared to Table 1).
  9. In line 234, it is stated that the samples "were manufactured in four groups of three internal diameters each of 109.8, 109.7, 109.6 and 109.5 mm", i.e. with a difference from the nominal value of 0.2 - 0.5 mm. So how were the specified tolerances, which are listed above, met?
  10. In lines 237-238 the authors state "The machining was carried out with a cutting speed of 100 m/min, a feed of 0.3 mm/rev and depth of cut 0.5 mm, in two passes without cooling. ...", but in the introduction to the Materials and Methods chapter (lines 120 - 125) the values ​​of the production parameters are completely different: ap = 0.3 mm; f = 0.2 mm/rev; vc = 300 m/min.
  11. Please justify, explain, and set things straight, because this makes the research and its results seem unreliable and irrelevant.
  12. Figure 5 - graphs - I am not sure about the measurements - was the measurement of the compressive load done until the sample was damaged? Displacements (deformations) of 23 mm, resp. 26 mm were measured Fig. 6, for samples with a size of 30 mm?
  13. Explain the sudden and sharp changes in the recorded forces, as well as the fluctuation of the recorded force values.
  14. Conduct a deeper discussion with regard to the recorded force-strain relationships and their significance, as well as the significance of changes in the behavior of the samples.
  15. Conduct a deeper discussion on the suitability of the selected material for dynamic loading, especially with regard to wear and service life. Consider (also discuss) other influences - temperature influence, chemical stability/resistance, viscosity/possible lubrication, the influence of the dusty environment in the mine, humidity, etc.
  16. Were the results for identical samples statistically processed? If yes, describe the methods and conclusions, if not, justify why.

Author Response

1. The authors in the Abstract talk about "determining the optimal diameter of the bore of bearing bushings made of a polymer designated as PA6"...

Response: It seems like an oversight made by reviewer, paper deals with optimizing PA6 housing, not bushing (yellow shaded in abstract’ sentence). PS In addition, small ‘h’ and ‘b’ look alike in document using Palatino Linotype font.

2. What do the authors consider optimal, what criteria must the optimal diameter meet and what method was used to perform the optimization?

Response: In this context “optimal” housing dimensions are those that enable multiple rolling bearing replacement process without compromising the operational performance of the roller in general. Authors understand that term “optimal” might be inappropriate or problematic regarding study’s aim, so changed it to “suitable”. We found it most appropriate for this context.

3. Lines 120 - 125 - On what basis were the specified machining parameters used in the given research? Note: surface roughness is a consequence of the machining parameter settings, it does not belong to the settings.

Response: Suggestion accepted. The authors fully agree with the reviewer's comment and express their gratitude for pointing out this issue. It’s was simply an oversight, since surface roughness can be only a consequence of machining process. Changes are introduced in main text. In addition, a forementioned parameters are empirical, which depend on the cutting tool, the machine they are working on, and the work piece material.

4. Lines 115 - 117, similarly 125 - 126, - references to literary sources are missing in the statements.

Responses: Sources/references have been added in both cases in the main text, along with the following sentence and thorough explanation in Section 2 on the statement regarding PA6 application temperature limits:

“Along with the use of standard lithium-based grease for bearings (such as SKF LGMT2 grease [23]), PA6 could be used for application at temperature of 120 °C (for a shorter period even up to 160 °C), with a long-term operational time at 100 °C [24] without changes of its mechanical properties.”

5. The designations in the text of symbols and variables should be the same as in the equations (slanted, with subscripts, etc ...) e.g. line 187 pmin, line 195 d; but it is necessary to check this in the entire manuscript.

Response: Suggestion accepted. Once again authors apologize for obvious typos (due to most probably auto-correction) and oversight. We double checked whole manuscript and each symbols. Corrections have been introduced in the main text.

6. I see a question mark after equation (5), please check.

Response: Authors double checked this, and no question mark has been observed (in .doc and .pdf as well).

7. Justify why static friction coefficient of µk = 0.3 was adopted for the subsequent calculations.

Response: Authors fully agree on the need for justification of such action: lower value of static friction coefficient  (0.03) was taken as a precaution measure, i.e. safety factor. By literature (Mitrović R, Ristivojević M, Rosić B.. Mašinski Elementi 1, University of Belgrade, Faculty of Mechanical Engineering, Belgrade 2019; table P3-4, μ should be 0.33). Following sentence is added in the main text, Section 3:
“As a justification, lower value of static friction coefficient µs was adopted for (higher) safety factor considerations.”

8. In lines 197 and 198, the authors talk about tolerances of 35 um, but in Table 1 about tolerances with a value of 0.035 um. What technology do the authors want to use to produce a 110 mm diameter component with a tolerance of 0.015 or 0.035 um? Water jet? Below in line 201, the recommended tolerance value is 15 um (this is a big difference compared to Table 1).

Response: Once again authors apologize for obvious typos – it should stand mm instead of µm. Authors double checked whole manuscript, each symbol and each equation. Corrections had been in one equation and few spots in the main text.

In addition, introduction of Section 4 has been rewritten and reshaped with thorough explanation on manufacturing technique of housing samples.

9. In line 234, it is stated that the samples "were manufactured in four groups of three internal diameters each of 109.8, 109.7, 109.6 and 109.5 mm", i.e. with a difference from the nominal value of 0.2 - 0.5 mm. So how were the specified tolerances, which are listed above, met?

Response: Authors agree that clarification is needed on this matter (in general) along with table 3 results. The achieved dimensions of the housing bore deviate from the targeted ones due to manufacturing inaccuracy, specifically in relation to the Average interference fit in class, which can be seen in the table 2. Comment on this matter is provided below table 2, as a following sentence:

“Results shown in table 2 indicate that the achieved dimensions of the housing bore deviate from the targeted ones due to manufacturing inaccuracy, resulting in the difference between the target interference fit and the average interference fit values in class.

10. In lines 237-238 the authors state "The machining was carried out with a cutting speed of 100 m/min, a feed of 0.3 mm/rev and depth of cut 0.5 mm, in two passes without cooling. ...", but in the introduction to the Materials and Methods chapter (lines 120 - 125) the values ​​of the production parameters are completely different: ap = 0.3 mm; f = 0.2 mm/rev; vc = 300 m/min.

Response: Once again authors apologize for obvious mistake – machining parameters were the ones given in the Section 2. Materials & Methods. Disputed sentence has been rewritten.

11. Please justify, explain, and set things straight, because this makes the research and its results seem unreliable and irrelevant.

Response: We fully agree with reviewer’s comment – whole manuscript has been reshaped and reordered, with one new chapter added, resulting splitting Section 4 into 2 new ones, dedicated to 1) used methods and equipment and 2) results and analysis.

12. Figure 5 - graphs - I am not sure about the measurements - was the measurement of the compressive load done until the sample was damaged? Displacements (deformations) of 23 mm, resp. 26 mm were measured Fig. 6, for samples with a size of 30 mm?

Response: Authors fully understand that our lack of precise and thorough description led to this misunderstanding and wrong interpretation. In this particular case, displacement is not the deformation - it represents displacement measure in millimeters during pressing and dismantling process (measured by sensor); accurately displacement represents relative position of the roller bearing and the housing at the beginning of the pressing-in / dismantling process itself. The following sentence has been added in the main text to resolve this problem:

“In this particular case measured displacement on each diagram represents the relative position of the roller bearing and the housing at the beginning of the pressing-in / dismantling process (when the displacement equals 0) and at the end of the force measurement process.”

13. Explain the sudden and sharp changes in the recorded forces, as well as the fluctuation of the recorded force values.

Response: Suggestion accepted. Following paragraph is added in the main text:

As a general observation, sudden transient ‘jumps’ and ‘drops’ (i.e. fluctuations) in the force values (for instance in Fig. 6, partly in Fig. 7) can be primarily attributed to insufficient interference fit between rolling bearing and PA6 housing, which is clearly evident in case of PA6 test specimens 102, 202, and 302, and partially in 103. In addition, force ‘drops’ could be as well attributed to the non-uniformity of the surface texture. For the remaining specimens, the force increments observed during both press-fitting and dismantling processes were in direct correlation with the recoverable elastic strain. As a recall, it is essential to emphasize that the displacement of the pressing machine was conducted in small intermittently steps every few seconds, i.e. after the stabilizing the force value on the sensor. In this context, this provides (as well) a justification for the selection of the static friction coefficient in the preliminary analytical calculation.”

14. Conduct a deeper discussion with regard to the recorded force-strain relationships and their significance, as well as the significance of changes in the behavior of the samples.

Response: Suggestion accepted. Paragraph from previous bullet partly covers this comment as well.

15. Conduct a deeper discussion on the suitability of the selected material for dynamic loading, especially with regard to wear and service life. Consider (also discuss) other influences - temperature influence, chemical stability/resistance, viscosity/possible lubrication, the influence of the dusty environment in the mine, humidity, etc.

Response: Unfortunately, authors of this study were not in technically possibility to conduct suggested tests, although some of them were listed to be next step(s) regarding investigation of PA6 housing. Authors appreciate the reviewer's good suggestion on this matter.

16. Were the results for identical samples statistically processed? If yes, describe the methods and conclusions, if not, justify why.

Response: No, we did not perform it. Authors considered that sample size is too small, thus it led to the conclusion that any form of statistical processing (or analysis) would not be relevant.

Reviewer 2 Report

Comments and Suggestions for Authors
  1. “In the last paragraph of Abstract”, summarize the idea of the article, and prepare what methods to improve the idea, and what tests to verify.
  2. It is recommended to modify the whole “2. Materials and Methods” e.g., There is no need to introduce PA56 if it is purchased from the manufacturer, if it is prepared by the method.
  3. Testing methods and means need to be given separately, rather than in the analysis, e.g.,  “This investigation began with examination of the surface of each housing sample made of PA6. The surface analysis was performed using a Hirox KH 7700 digital microscope (HYROX, Japan, 2016).”
  4. Figure 2 needs to be further processed; the scale and image contrast are not obvious.
  5. The overall layout of the article is problematic, and it is suggested to revise the whole, especially the layout of the figure, which is not beautiful.
  6. Do not give Figure 3 separately; it is recommended to merge the relevant data maps.
  7. Figure 6 is poorly processed, and no error correction is made.

Comments on the Quality of English Language
  1. The introduction of literature in the article is less and not standardized.

Author Response

1. “In the last paragraph of Abstract”, summarize the idea of the article, and prepare what methods to improve the idea, and what tests to verify.

Response: Suggestion accepted. As a consequence, authors consider that whole abstract needs to be rewritten and rephrased. New abstract is:

This paper presents an experimental method for determining the suitable bore diameter of bearing housings made of polymer designated as PA6, which enables multiple bearing replacement processes. Preceded by analytical calculation, four distinct series of housing samples (each with varying production tolerances) were subjected to testing, where each series comprised three housing samples with identical tolerance specifications. The assembly and disassembly processes of press-fit joints were thoroughly monitored using a force sensor, complemented by equipment for measuring the roughness of contact surfaces. Based on the experimental findings, a recommendation is provided for an appropriate interference fit for the tested bearing housing providing suitable solution for multiple maintenance process. As a summary, the idea of this research is to define the prototype solution for the interference fit of a rolling bearing installed in PA6 housing. Methods used to examine proposed solution were surface topography and roundness measuring of PA6 housings, while the press-fitting and dismantling tests of rolling bearings in/from PA6 housings were used to verify it.

2. It is recommended to modify the whole “2. Materials and Methods” e.g., There is no need to introduce PA56 if it is purchased from the manufacturer, if it is prepared by the method.

Response: Suggestion partly accepted. Authors consider that PA6 (in this particular case) as a semi-finished product, from which housings were made, deserve to be described in one small paragraph at least. Section 2 has been slightly changed according to the this and one of reviewer 1’s comment as well.

3. Testing methods and means need to be given separately, rather than in the analysis, e.g.,  “This investigation began with examination of the surface of each housing sample made of PA6. The surface analysis was performed using a Hirox KH 7700 digital microscope (HYROX, Japan, 2016).”

Response: Suggestion accepted. Suggestion is very constructive leading the splitting previously Section 4. Experimental testing into two new ones: 4. Experiment methods and 5. Results and analysis. Section 5. has been expanded with appropriate analysis on changes in the recorded forces, and fluctuation of the recorded force values (for instance). Authors think that this splitting would increase the readability and emphasize the contributions of this research.

4. Figure 2 needs to be further processed; the scale and image contrast are not obvious.

Response: Suggestion accepted. Figure 2 has been replaced with the new one with higher resolution and better quality.

5. The overall layout of the article is problematic, and it is suggested to revise the whole, especially the layout of the figure, which is not beautiful.

Response: Suggestion accepted. The manuscript has been substantially reshaped and enhanced, with two new Sections introduced. Figures have been enhanced as well, with new labels clarifying what they represent.

6. Do not give Figure 3 separately; it is recommended to merge the relevant data maps.

Response: Suggestion accepted. In addition, Figure 3a has been moved to (newly) introduced Section 4. Testing methods, while Figures 3b-e (now a and d) has been moved to (newly) introduced Section 5. Results and analysis.

7. Figure 6 is poorly processed, and no error correction is made.

Response: Suggestion accepted. Figures depicting force-displacement diagrams (for each sample) during each cycles of press-fitting and dismantling process have been replaced with new ones, clearly labeled with clear indication on samples designation (#) .

Round 2

Reviewer 1 Report

Comments and Suggestions for Authors

The authors responded adequately, the manuscript is publishable in its current state.